# Adversarially Robust Few-Shot Learning: A Meta-Learning Approach

**Micah Goldblum**[*]
Department of Mathematics
University of Maryland
goldblum@umd.edu

**Liam Fowl**[*]
Department of Mathematics
University of Maryland
lfowl@umd.edu

**Tom Goldstein**
Department of Computer Science
University of Maryland
tomg@cs.umd.edu

## Abstract

Previous work on adversarially robust neural networks for image classification requires large training sets and computationally expensive training procedures. On the other hand, few-shot learning methods are highly vulnerable to adversarial examples. The goal of our work is to produce networks which both perform well at few-shot classification tasks and are simultaneously robust to adversarial examples. We develop an algorithm, called Adversarial Querying (AQ), for producing adversarially robust meta-learners, and we thoroughly investigate the causes for adversarial vulnerability. Moreover, our method achieves far superior robust performance on few-shot image classification tasks, such as Mini-ImageNet and CIFAR-FS, than robust transfer learning.

## 1 Introduction

For safety-critical applications like facial recognition, algorithmic trading, and copyright control, adversarial attacks pose an actionable threat [35, 12, 25]. Conventional adversarial training and pre-processing defenses aim to produce networks that resist attack [19, 34, 26], but such defenses rely heavily on the availability of large training datasets. In applications that require *few-shot learning*, such as face recognition from few images, recognition of a video source from a single clip, or recognition of a new object from few example photos, the conventional robust training pipeline breaks down.

When data is scarce or new classes arise frequently, neural networks must adapt quickly [7, 14, 24, 30]. In these situations, *meta-learning* methods conduct few-shot learning by creating networks that learn quickly from little data and with computationally cheap fine-tuning. While state-of-the-art meta-learning methods perform well on benchmark few-shot classification tasks, these naturally trained neural networks are highly vulnerable to adversarial examples. In fact, even adversarially trained feature extractors fail to resist attacks in the few-shot setting (see Section 4.1).

We propose a new approach, called *adversarial querying*, in which the network is exposed to adversarial attacks during the query step of meta-learning. This algorithm-agnostic method produces a feature extractor that is robust, even without adversarial training during fine-tuning. In the few-shot setting, we show that adversarial querying outperforms other robustness techniques by a wide margin in terms of both clean accuracy and adversarial robustness (see Table 1). We solve the following minimax problem:

$$\min_{\theta} \mathbb{E}_{S,(\mathbf{x},y)} \left[ \max_{\|\delta\|_p < \epsilon} \mathcal{L}(F_{A(\theta,S)}, \mathbf{x} + \delta, y) \right], \tag{1}$$

where $S$ and $(\mathbf{x}, y)$ are data sampled from the training distribution, $A$ is a fine-tuning algorithm for the model parameters, $\theta$, and $\epsilon$ is a $p$-norm bound for the attacker. In Section 4, we further

---

[*]Authors contributed equally.

motivate adversarial querying and exhibit a wide range of experiments. To motivate the necessity for adversarial querying, we test methods, such as adversarial fine-tuning and pre-processing defenses, which if successful, would eliminate the need for expensive adversarial training routines. We find that these methods are far less effective than adversarial querying.

| Model | $\mathcal{A}_{nat}$ | $\mathcal{A}_{adv}$ |
|---|---|---|
| Naturally Trained R2-D2 | 73.01 % ± 0.13 | 0.00 % ± 0.13 |
| AT transfer (R2-D2 backbone) | 39.13 % ± 0.13 | 25.33% ± 0.13 |
| ADML | 47.75% ± 0.13 | 18.49 % ± 0.13 |
| AQ R2-D2 (ours) | 57.87% ± 0.13 | **31.52%** ± 0.13 |

Table 1: R2-D2 [3], adversarially trained transfer learning, ADML [33], and our adversarially queried (AQ) R2-D2 model on 5-shot Mini-ImageNet. Natural accuracy is denoted $\mathcal{A}_{nat}$, and robust accuracy, $\mathcal{A}_{adv}$, is computed with a 20-step PGD attack as in [19] with $\epsilon = 8/255$. A description of our training regime can be found in Appendix A.5. All results are tested on $150000$ total samples, so confidence intervals of one standard error are of width at most $100\sqrt{(0.5)(0.5)/150000}\% < 0.13\%$.

## 2 Related Work

### 2.1 Learning with less data

Before the emergence of meta-learning, a number of approaches existed to cope with few-shot problems. One simple approach is *transfer learning*, in which pre-trained feature extractors are trained on large data sets and fine-tuned on new tasks [2, 11]. Metric learning methods avoid overfitting to the small number of training examples in new classes by instead performing classification using nearest-neighbors in feature space with a feature extractor that is trained on a large corpus of data and not re-trained when classes are added [29, 9, 20]. Metric learning methods are computationally efficient when adding new classes at inference, since the feature extractor is not re-trained.

Meta-learning algorithms create a "base" model that quickly adapts to new tasks by fine-tuning. This model is created using a set of training tasks $\{\mathcal{T}_i\}$ that can be sampled from a task distribution. Each task comes with *support* data, $\mathcal{T}_i^s$, and *query* data, $\mathcal{T}_i^q$. Support data is used for fine-tuning, and query data is used to measure the performance of the resulting network. In practice, each task is taken to be a classification problem involving only a small subset of classes in a large many-class data set. The number of examples per class in the support set is called the *shot*, so that fine-tuning on five support examples per class is 5-shot learning.

An iteration of training begins by sampling tasks $\{\mathcal{T}_i\}$ from the task distribution. In the "inner loop" of training, the base model is fine-tuned on the support data from the sampled tasks. In the "outer loop" of training, the fine-tuned network is used to make predictions on the query data, and the base model parameters are updated to improve the accuracy of the resulting fine-tuned model. The outer loop requires backpropagation through the fine-tuning steps. A formal treatment of the prototypical meta-learning routine can be found in Algorithm 1.

---

**Algorithm 1** The meta-learning framework

---

**Require:** Base model, $F_\theta$, fine-tuning algorithm, $A$, learning rate, $\gamma$, and distribution over tasks, $p(\mathcal{T})$.
Initialize $\theta$, the weights of $F$;
**while** not done **do**
    Sample batch of tasks, $\{\mathcal{T}_i\}_{i=1}^n$, where $\mathcal{T}_i \sim p(\mathcal{T})$ and $\mathcal{T}_i = (\mathcal{T}_i^s, \mathcal{T}_i^q)$.
    **for** $i = 1, \ldots, n$ **do**
        Fine-tune model on $\mathcal{T}_i$ (inner loop). New network parameters are written $\theta_i = A(\theta, \mathcal{T}_i^s)$.
        Compute gradient $g_i = \nabla_\theta \mathcal{L}(F_{\theta_i}, \mathcal{T}_i^q)$
    **end for**
    Update base model parameters (outer loop):
    $\theta \leftarrow \theta - \frac{\gamma}{n} \sum_i g_i$
**end while**

---

Note that the fine-tuned parameters, $\theta_i = A(\theta, \mathcal{T}_i^s)$, in Algorithm 1, are a function of the base model's parameters so that the gradient computation in the outer loop may backpropagate through $A$. For validation after training, the base model is fine-tuned on the support set of hold-out tasks, and accuracy on the query set is reported. In this work, we report performance on Omniglot, Mini-ImageNet, and CIFAR-FS [16, 31, 3].

We focus on four meta-learning algorithms: MAML, R2-D2, MetaOptNet, and ProtoNet [8, 3, 18, 29]. During fine-tuning, MAML uses SGD to update all parameters, minimizing cross-entropy loss. Since unrolling SGD steps into a deep computation graph is expensive, first-order variants have been developed to avoid computing second-order derivatives. We use the original MAML formulation. R2-D2 and MetaOptNet, on the other hand, only update the final linear layer during fine-tuning, leaving the "backbone network" that extracts these features frozen at test time. R2-D2 replaces SGD with a closed-form differentiable solver for regularized ridge regression, while MetaOptNet achieves its best performance when replacing SGD with a solver for SVM. Because the objective of these linear problems is convex, differentiable convex optimizers can be efficiently deployed to find optima, and differentiate these optima with respect to the backbone parameters at train time. ProtoNet takes an approach inspired by metric learning. It constructs class prototypes as centroids in feature space for each task. These centroids are then used to classify the query set in the outer loop of training. Because each class prototype is a simple geometric average of feature representations, it is easy to differentiate through the fine-tuning step.

## 2.2 Adversarial attacks and defenses

Following standard practices, we assess the robustness of models by attacking them with $\ell_\infty$-bounded perturbations. We craft adversarial perturbations using the projected gradient descent attack (PGD) since it has proven to be one of the most effective algorithms both for attacking as well as for adversarial training [19]. A detailed description of the PGD attack algorithm can be found in Appendix A.9. We consider perturbations with $\ell_\infty$ bound $8/255$ and a step size of $2/255$ as described by [19]. *Adversarial training* is the industry standard for creating robust models that maintain good clean-label performance [19]. This method involves replacing clean examples with adversarial examples during the training routine. A simple way to harden models to attack is adversarial training, which solves the minimax problem

$$\min_\theta \mathbb{E}_{(\mathbf{x},y)\sim\mathcal{D}} \left[ \max_{\|\delta\|_p < \epsilon} \mathcal{L}_\theta(\mathbf{x} + \delta, y) \right], \tag{2}$$

where $\mathcal{L}_\theta(\mathbf{x} + \delta, y)$ is the loss function of a network with parameters $\theta$, $\mathbf{x}$ is an input image with label $y$, and $\delta$ is an adversarial perturbation. Adversarial training finds network parameters which maintain low loss (and correct class labels) even when adversarial perturbations are added. A number of adversarial training variants have emerged which improve performance or achieve various tasks from data augmentation to domain generalization to model compression [28, 10, 23].

## 2.3 Robust learning with less data

Several authors have tried to learn robust models in the data scarce regime. The authors of [27] study robustness properties of transfer learning. They find that retraining earlier layers of the network during fine-tuning impairs the robustness of the network, while only retraining later layers can largely preserve robustness. ADML is the first attempt at achieving robustness through meta-learning. ADML is a MAML variant specifically designed specifically for robustness [33]. However, this method for robustness is designed only for MAML, an algorithm which is now far from state-of-the-art. Moreover, ADML is computationally expensive, and the authors only test their method against a weak attacker. We re-implement ADML and test it against a strong attacker. We show that our method simultaneously achieves higher robustness and higher natural accuracy.

## 3 Naturally trained meta-learning methods are not robust

In this section, we benchmark the robustness of existing meta-learning methods. Similarly to classically trained classifiers, we expect that few-shot learners are highly vulnerable to attack when adversarial defenses are not employed. We test prominent meta-learning algorithms against a 20-step PGD attack as in [19]. Table 2 contains natural and robust accuracy on the Mini-ImageNet and

CIFAR-FS 5-shot tasks [31, 3]. Experiments in the 1-shot setting can be found in Appendix A.1. All results are tested on $150000$ total samples, so confidence intervals of one standard error are of width at most $100\sqrt{\frac{(0.5)(0.5)}{150000}}\% < 0.13\%$.

| Model | $\mathcal{A}_{nat}$ MI | $\mathcal{A}_{adv}$ MI | $\mathcal{A}_{nat}$ FS | $\mathcal{A}_{adv}$ FS |
|---|---|---|---|---|
| ProtoNet | 70.23% | 0.00% | 79.66% | 0.00% |
| R2-D2 | 73.02% | 0.00% | 82.81% | 0.00% |
| MetaOptNet | 78.12% | 0.00% | 84.11% | 0.00% |

Table 2: 5-shot MiniImageNet (MI) and CIFAR-FS (FS) results comparing naturally trained meta-learners. $\mathcal{A}_{nat}$ and $\mathcal{A}_{adv}$ are natural and robust test accuracy, respectively, where robust accuracy is computed with respect to a 20-step PGD attack.

We find that these algorithms are completely unable to resist the attack. Interestingly, MetaOptNet uses SVM for fine-tuning, which is endowed with a wide margins property. The failure of even SVM to express robustness during testing suggests that using robust fine-tuning methods (at test time) on naturally trained meta-learners is insufficient for robust performance. To further examine this, we consider MAML, which updates the entire network during fine-tuning. We use a naturally trained MAML model and perform adversarial training during fine-tuning (see Table 3). Adversarial training is performed with 7-PGD as in [19]. If adversarial fine-tuning yielded robust classification, then we could avoid expensive adversarial training variants during meta-learning.

| Model | $\mathcal{A}_{adv}$ | $\mathcal{A}_{adv(AT)}$ |
|---|---|---|
| 1-shot Mini-ImageNet | 0.03% | 0.20% |
| 5-shot Mini-ImageNet | 0.03% | 1.55% |
| 1-shot Omniglot | 68.46% | 74.66% |
| 5-shot Omniglot | 82.28% | 87.94% |

Table 3: MAML models on Mini-ImageNet and Omniglot. $\mathcal{A}_{nat}$ and $\mathcal{A}_{adv}$ are natural and robust test accuracy, respectively, where robust accuracy is computed with respect to a 20-step PGD attack. $\mathcal{A}_{nat(AT)}$ and $\mathcal{A}_{adv(AT)}$ are natural and robust test accuracy with 7-PGD fine-tuning.

While clean trained MAML models with adversarial fine-tuning are slightly more robust than their naturally fine-tuned counterparts, they achieve almost no robustness on Mini-ImageNet. Omniglot is an easier data set, and the performance of adversarially fine-tuned MAML on the 5-shot version is below a reasonable tolerance for robustness. We conclude from these experiments that naturally trained meta-learners are vulnerable to adversarial examples, and robustness techniques specifically for few-shot learning are required.

## 4 Adversarial querying: a meta-learning algorithm for adversarial robustness

We now introduce adversarial querying (AQ), an adversarial training algorithm for meta-learning. Let $A(\theta, S)$ denote a fine-tuning algorithm. Then, $A$ is a map from support data set, $S$, and network parameters, $\theta$, to parameters for the fine-tuned network. Then, we seek to solve the following minimax problem (Equation 1 revisited):

$$\min_{\theta} \mathbb{E}_{S,(\mathbf{x},y)} \left[ \max_{\|\delta\|_p < \epsilon} \mathcal{L}(F_{A(\theta,S)}, \mathbf{x} + \delta, y) \right],$$

where $S$ and $(\mathbf{x}, y)$ are support and query data, respectively, sampled from the training distribution, and $\epsilon$ is a $p$-norm bound for the attacker. Thus, the objective is to find a central parameter vector which, when fine-tuned on support data, minimizes the expected query loss against an attacker. We approach this minimax problem with an alternating algorithm consisting of the following steps:

1. Sample support and query data
2. Fine-tune on the support data (inner loop)

3. Perturb query data to maximize loss
4. Minimize query loss, backpropagating through the fine-tuning algorithm (outer loop)

A formal treatment of this method is presented in Algorithm 2. Adversarial querying requires a factor of $n + 1$ more SGD steps than standard meta-learning. We test adversarial querying across multiple data sets and meta-learning protocols. It is important to note that adversarial querying is algorithm-agnostic. We test this method on the ProtoNet, R2-D2, and MetaOptNet algorithms on CIFAR-FS and Mini-ImageNet (see Table 4 and Table 5). See Appendix A.6 for tests against additional attacks. We test against black-box transfer attacks which have been shown to be effective against gradient masking. In the white-box setting, we test adversarial querying against several gradient-based attacks as these have been shown to be more effective than zeroth order methods [4].

---

**Algorithm 2** Adversarial Querying

**Require:** Base model, $F_\theta$, fine-tuning algorithm, $A$, learning rate, $\gamma$, and distribution over tasks, $p(\mathcal{T})$.
Initialize $\theta$, the weights of $F$;
**while** not done **do**
    Sample batch of tasks, $\{\mathcal{T}_i\}_{i=1}^n$, where $\mathcal{T}_i \sim p(\mathcal{T})$ and $\mathcal{T}_i = (\mathcal{T}_i^s, \mathcal{T}_i^q)$.
    **for** $i = 1, ..., n$ **do**
        Fine-tune model on $\mathcal{T}_i$. New network parameters are written $\theta_i = A(\theta, \mathcal{T}_i^s)$.
        Construct adversarial query data, $\widehat{\mathcal{T}_i^q}$, by maximizing $\mathcal{L}(F_{\theta_i}, \widehat{\mathcal{T}_i^q})$ constrained to $\|\widehat{\mathbf{x}_j^q} - \mathbf{x}_j^q\|_p <$
        $\epsilon$ for query examples, $\mathbf{x}_j^q$, and their associated adversaries, $\widehat{\mathbf{x}_j^q}$.
        Compute gradient $g_i = \nabla_\theta \mathcal{L}(F_{\theta_i}, \widehat{\mathcal{T}_i^q})$.
    **end for**
    Update base model parameters: $\theta \leftarrow \theta - \frac{\gamma}{n} \sum_i g_i$
**end while**

---

| Model | $\mathcal{A}_{nat}$ 1-Shot | $\mathcal{A}_{adv}$ 1-Shot | $\mathcal{A}_{nat}$ 5-Shot | $\mathcal{A}_{adv}$ 5-Shot |
|---|---|---|---|---|
| ProtoNet AQ | 42.33% | 26.48% | 63.53% | 40.11% |
| R2-D2 AQ | 52.38% | **32.33%** | 69.25% | **44.80%** |
| MetaOptNet AQ | 53.27% | 30.74% | 71.07% | 43.79% |

Table 4: Comparison of adversarially queried (AQ) meta-learners on 1-shot and 5-shot CIFAR-FS. $\mathcal{A}_{nat}$ and $\mathcal{A}_{adv}$ are natural and robust test accuracy, respectively, where robust accuracy is computed with respect to a 20-step PGD attack. Top 1-shot and 5-shot robust accuracy is bolded.

| Model | $\mathcal{A}_{nat}$ 1-Shot | $\mathcal{A}_{adv}$ 1-Shot | $\mathcal{A}_{nat}$ 5-Shot | $\mathcal{A}_{adv}$ 5-Shot |
|---|---|---|---|---|
| ProtoNet AQ | 33.31% | 17.69% | 52.04% | 27.99% |
| R2-D2 AQ | 37.91% | **20.59%** | 57.87% | **31.52%** |
| MetaOptNet AQ | 43.74% | 18.37% | 60.71% | 28.08% |

Table 5: Comparison of adversarially queried (AQ) meta-learners on 1-shot and 5-shot Mini-ImageNet. $\mathcal{A}_{nat}$ and $\mathcal{A}_{adv}$ are natural and robust test accuracy, respectively, where robust accuracy is computed with respect to a 20-step PGD attack. Top 1-shot and 5-shot robust accuracy is bolded.

In our tests, R2-D2 outperforms MetaOptNet in robust accuracy despite having a less powerful backbone architecture. In Section 4.5, we dissect the effects of backbone architecture and classification head on robustness of meta-learned models. In Appendix A.6, we verify that adversarial querying generates networks robust to a wide array of strong attacks.

### 4.1 Adversarial querying is more robust than transfer learning from adversarially trained models

We observe above that few-shot learning methods with a non-robust feature extractor break under attack. But what if we use a robust feature extractor? In the following section, we consider both transfer learning and meta-learning with a robust feature extractor.

In order to compare robust transfer learning and meta-learning, we train the backbone networks from meta-learning algorithms on all training data simultaneously in the fashion of standard adversarial training using 7-PGD (not meta-learning). We then fine-tune using the head from a meta-learning algorithm on top of the transferred feature extractor. We compare the performance of these feature extractors to that of those trained using adversarially queried meta-learning algorithms with the same backbones and heads. This experiment provides a direct comparison of feature extractors produced by robust transfer learning and robust meta-learning (see Table 6). Meta-learning exhibits far superior robustness than transfer learning for all algorithms we test. Additional experiments on CIFAR-FS and on 1-shot Mini-ImageNet can be found in Appendix A.2.

| Model | $\mathcal{A}_{nat}$ Transfer | $\mathcal{A}_{adv}$ Transfer | $\mathcal{A}_{nat}$ Meta | $\mathcal{A}_{adv}$ Meta |
|---|---|---|---|---|
| MAML | 32.79% | 18.03% | **33.45%** | **23.07%** |
| ProtoNet | 31.14% | 22.31% | **52.04%** | **27.99%** |
| R2-D2 | 39.13% | 25.33% | **57.87%** | **31.52%** |
| MetaOptNet | 50.23% | 22.45% | **60.71%** | **28.08%** |

Table 6: Adversarially trained transfer learning and adversarially queried meta-learning on 5-shot Mini-ImageNet. $\mathcal{A}_{nat}$ and $\mathcal{A}_{adv}$ are natural and robust test accuracy, respectively, where robust accuracy is computed with respect to a 20-step PGD attack. Top natural and robust accuracy for each architecture is bolded.

## 4.2 Why attack only query data?

In the adversarial querying procedure detailed in Algorithm 2, we only attack query data. Consider that the loss value on query data represents performance on testing data after fine-tuning on the support data. Thus, low loss on perturbed query data represents robust accuracy on testing data after fine-tuning. Simply put, minimizing loss on adversarial query data moves the parameter vector towards a network with high robust test accuracy. It follows that attacking only support data is not an option for achieving robust meta-learners. Attacking support data but not query data can be seen as maximizing clean test accuracy when fine-tuned in a robust manner, but since we want to maximize *robust* test accuracy, this would be inappropriate. One question remains: should we attack both query and support data?

One reason to perturb only query data is computational efficiency. The bulk of computation in adversarial training is spent computing adversarial attacks since perturbing each batch requires an iterative algorithm. Perturbing support data doubles the number of adversarial attacks computed during training. Thus, only attacking query data significantly accelerates training. Perturbing support data will additionally increase the cost of fine-tuning during deployment, especially for methods, such as MetaOptNet, which use efficient solvers for the linear classification problem on clean data. But if attacking support data during the inner loop of training were to significantly improve robust performance, we would like to know.

We now compare adversarial querying to a variant in which support data is also perturbed during training. We use MAML to conduct this comparison on the Omniglot and Mini-ImageNet data sets. Additional experiments on 1-shot tasks can be found in Appendix A.3.

| Model | $\mathcal{A}_{nat}$ | $\mathcal{A}_{adv}$ | $\mathcal{A}_{nat(AT)}$ | $\mathcal{A}_{adv(AT)}$ |
|---|---|---|---|---|
| MAML (naturally trained) | 60.25% | 0.03% | 32.45% | 1.55% |
| MAML adv. query | 33.45% | **23.07%** | 33.03% | **23.29%** |
| MAML adv. query and support | 29.98% | 22.55% | 30.44% | 23.03% |
| ADML | 47.75% | 18.49% | 47.27% | 20.23% |

Table 7: Performance on 5-shot Mini-ImageNet. Robust accuracy, $\mathcal{A}_{adv}$, is computed with respect to a 20-step PGD attack. $\mathcal{A}_{nat(AT)}$ and $\mathcal{A}_{adv(AT)}$ are natural and robust test accuracy with 7-PGD training during fine-tuning. Top robust accuracy with and without adversarial fine-tuning is bolded.

In these experiments, we find that adversarially attacking the support data during the inner loop of meta-learning does not improve performance over adversarial querying. Furthermore, networks trained in this fashion require adversarial fine-tuning during test time or else they suffer a massive

| Model | $\mathcal{A}_{nat}$ | $\mathcal{A}_{adv}$ | $\mathcal{A}_{nat(AT)}$ | $\mathcal{A}_{adv(AT)}$ |
|---|---|---|---|---|
| MAML (naturally trained) | 97.12% | 82.28% | 97.71% | 87.94% |
| MAML adv. query | 97.27% | **95.85%** | 97.51% | **96.14%** |
| MAML adv. query and support | 95.61% | 77.73% | 97.46% | 95.65% |
| ADML | 97.31% | 94.19% | 97.56% | 94.82% |

Table 8: Performance on 5-shot Omniglot. Robust accuracy, $\mathcal{A}_{adv}$, is computed with respect to a 20-step PGD attack. $\mathcal{A}_{nat(AT)}$ and $\mathcal{A}_{adv(AT)}$ are natural and robust test accuracy with 7-PGD training during fine-tuning. Top robust accuracy with and without adversarial fine-tuning is bolded.

loss in robust test accuracy. Following these results and the significant reasons to avoid attacking support data, we subsequently only attack query data.

### 4.3 Tuning the robustness-accuracy trade-off: AQ may be used to adapt other robustness techniques to meta-learning

Adversarial querying trades off natural accuracy for robust accuracy. This massive trade-off exists in the standard setting where SOTA robust ImageNet models sacrifice twenty percentage points [32]. Adversarial querying can be used to construct meta-learning analogues for other variants of adversarial training. We explore this by using the TRADES loss function, used for controlling the accuracy-robustness trade-off, in the querying step of AQ [34]. We refer to this method as meta-TRADES. While meta-TRADES can marginally outperform our initial adversarial querying method in robust accuracy, we find that it trades off natural accuracy in the process. See Appendix A.4 for both 1-shot and 5-shot on multiple datasets.

### 4.4 For better natural and robust accuracy, only fine-tune the last layer.

High performing meta-learning models, like MetaOptNet and R2-D2, fix their feature extractor and only update their last linear layer during fine-tuning. In the setting of transfer learning, robustness is a feature of early convolutional layers, and re-training these early layers leads to a significant drop in robust test accuracy [27]. We verify that re-training only the last layer leads to improved natural and robust accuracy in adversarially queried meta-learners by training a MAML model but only updating the final fully-connected layer during fine-tuning including during the inner loop of meta-learning. We find that the model trained by only fine-tuning the last layer decisively outperforms the traditional MAML algorithm (AQ) in both natural and robust accuracy (see Table 9).

| Re-trained | $\mathcal{A}_{nat}$ | $\mathcal{A}_{adv}$ | $\mathcal{A}_{nat(AT)}$ | $\mathcal{A}_{adv(AT)}$ |
|---|---|---|---|---|
| All layers | 33.45% | 23.07% | 33.03% | 23.29% |
| FC only | **40.06%** | **25.15%** | **39.94%** | **25.32%** |

Table 9: Adversarially queried MAML compared with a MAML variant with only the last layer re-trained during fine-tuning on 5-shot Mini-ImageNet. $\mathcal{A}_{nat}$ and $\mathcal{A}_{adv}$ are natural and robust test accuracy, respectively, where robust accuracy is computed with respect to a 20-step PGD attack. $\mathcal{A}_{nat(AT)}$ and $\mathcal{A}_{adv(AT)}$ are natural and robust test accuracy, respectively with 7-PGD training during fine-tuning. Layers are fine-tuned for 10 steps with a learning rate of 0.01.

### 4.5 The R2-D2 head, not embedding, is responsible for superior robust performance.

The naturally trained MetaOptNet algorithm outperforms R2-D2 in natural accuracy, but previous research has found that performance discrepancies between meta-learning algorithms might be an artifact of different backbone networks [5]. We confirm that MetaOptNet with the R2-D2 backbone performs similarly to R2-D2 in the natural meta-learning setting (see Appendix Table 22).

However, we find that the performance discrepancy in the adversarial setting is *not* explained by differences in backbone architecture. In our adversarial querying experiments, we saw that MetaOptNet was less robust than R2-D2. This discrepancy remains when we train MetaOptNet with the R2-D2 backbone (see Appendix Table 21). We conclude that MetaOptNet's backbone is not responsible for its inferior robustness. These experiments suggest that ridge regression may

be a more effective fine-tuning technique than SVM for robust performance. ProtoNet with R2-D2 backbone also performs worse than the other two adversarially queried models with the same backbone architecture.

# 5  Robustness alternatives to adversarial training

## 5.1  Enhancing robustness with robust architectural features

In addition to adversarial training, architectural features have been used to enhance robustness [32]. Feature denoising blocks pair classical denoising operations with learned $1 \times 1$ convolutions to reduce the feature noise in feature maps at various stages of a network, and thus reduce the success of adversarial attacks. Massive architectures with these blocks have achieved state-of-the-art robustness against targeted adversarial attacks on ImageNet. However, when deployed on small networks for meta-learning, we find that denoising blocks do not improve robustness. We deploy denoising blocks identical to those in [32] after various layers of the R2-D2 network. The best results for the denoising experiments are achieved by adding a denoising block after the fourth layer in the R2-D2 embedding network (see Appendix Table 23).

## 5.2  Pre-processing defenses

Recent works have proposed pre-processing defenses for sanitizing adversarial examples before feeding them into a naturally trained classifier. If successful, these methods would avoid the expensive adversarial querying procedure during training. While this approach has found success in the mainstream literature, we find that it is ineffective in the few-shot regime.

In DefenseGAN, a GAN trained on natural images is used to sanitize an adversarial example by replacing (possible corrupted) test images with the nearest image in the output range of the GAN [26]. Unfortunately, GANs are not expressive enough to preserve the integrity of testing images on complex datasets involving high-res natural images, and recent attacks have critically compromised the performance of this defense [13, 1]. We found the expressiveness of the generator architecture used in the original DefenseGAN setup to be insufficient for even CIFAR-FS, so we substitute a stronger ProGAN generator to model the CIFAR-100 classes [15].

The supperesolution defense first denoises data with sparse wavelet filters and then performs super-resolution [22]. This defense is also motivated by the principle of projecting adversarial examples onto the natural image manifold. We test the superresolution defense using the same wavelet filtering and superresolution network (SRResNet) used by [22] and first introduced by [17]. Like with the generator for DefenseGAN, we train the SRResNet on the entire CIFAR-100 dataset before applying the superresolution defense.

We find that these methods are not well suited to the few-shot domain, in which the generative model or superresolution network may not be able to train on the little data available. Morever, even after training the generator on all CIFAR-100 classes, we find that DefenseGAN with a naturally trained R2-D2 meta-learner performs significantly worse in both natural and robust accuracy than an adversarially queried meta-learner of the same architecture. Similarly, the superresolution defense achieves little robustness. The results of these experiments can be found in Appendix Table 24.

# 6  Discussion

Naturally trained networks for few-shot image classification are vulnerable to adversarial attacks, and existing robust transfer learning methods do not perform well on few-shot tasks. Naturally trained networks suffer from adversarial vulnerability even when adversarially fine-tuned. We thus identify the need for few-shot methods for adversarial robustness. We particularly study robustness in the context of meta-learning. We develop an algorithm-agnostic method, called adversarial querying, for hardening meta-learning models. We find that meta-learning models are most robust when the feature extractor is fixed, and only the last layer is retrained during fine-tuning. We further identify that choice of classification head significantly impacts robustness. We believe that this paper is a starting point for developing adversarially robust methods for few-shot applications.

A PyTorch implementation of adversarial querying can be found at:
`https://github.com/goldblum/AdversarialQuerying`

## Broader Impact

Few-shot learning systems are already deployed in real-world settings, but practitioners may remain unaware of the robustness properties of their models. Our work thoroughly studies this topic using methods which these practitioners may deploy, and we contribute a method for hardening their systems. Our work can benefit both organizations deploying few-shot learning systems as well as their customers and clients.

## Funding Disclosure

This work was supported by the DARPA GARD and DARPA QED programs, in addition to the National Science Foundation Division of Mathematical Sciences.

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
