[Supplementary Material]

# A   Appendix

## A.1   Additional experiments testing the robustness of naturally trained meta-learning models

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

Table 16: 5-shot Mini-ImagNet (MI) and CIFAR-FS (FS) results comparing meta-TRADES to adversarial querying (AQ). All models are based on R2-D2. $\lambda$ is the parameter for TRADES loss. $\mathcal{A}_{nat}$ and $\mathcal{A}_{adv}$ are natural and robust test accuracy, respectively, where robust accuracy is computed with respect to a 20-step PGD attack.

| Model | $\mathcal{A}_{nat}$ MI | $\mathcal{A}_{adv}$ MI | $\mathcal{A}_{nat}$ FS | $\mathcal{A}_{adv}$ FS |
|---|---|---|---|---|
| R2-D2 AQ | 37.91% | 20.59% | 52.38% | 32.33% |
| R2-D2 TRADES ($1/\lambda = 1$) | 39.11% | 20.25% | 48.77% | 31.99% |
| R2-D2 TRADES ($1/\lambda = 6$) | 34.27% | 22.00% | 44.37% | 33.55% |

Table 17: 1-shot Mini-ImagNet (MI) and CIFAR-FS (FS) results comparing meta-TRADES to adversarial querying. $\mathcal{A}_{nat}$ and $\mathcal{A}_{adv}$ are natural and robust test accuracy, respectively, where robust accuracy is computed with respect to a 20-step PGD attack.

## A.5 Training hyperparameters

We train ProtoNet, R2-D2, and MetaOptNet models for 60 epochs with SGD. We use a learning rate of $0.1$, momentum (Nesterov) of $0.9$, and a weight decay term of $5(10^{-4})$ for the parameters of both the head and the embedding. We decrease the learning rate to $0.06$ after epoch 20, $0.012$ after epoch 40, and $0.0024$ after epoch 50. MAML is trained for 60000 epochs with meta learning rate of $0.001$ and fine-tuning learning rate of $0.01$. Fine-tuning is performed for 10 steps per task. We did not perform a hyperparameter search and combined common hyperparameters for PGD training with meta-learning hyperparameters used for MetaOptNet and MAML. Experiments for this paper are performed on a machine with $4\times$ NVIDIA RTX 2080 Ti graphics cards. Runtime comparisons can be found in Table 25.

## A.6 Resistance to other attacks

We test our method by exposing our adversarially queried R2-D2 model to a variety of powerful adversarial attacks. We implement the momentum iterated fast gradient sign method (MI-FGSM), DeepFool, and 20-step PGD with 20 random restarts [6, 21, 19]. Our adversarially queried model indeed is nearly as robust against the strongest $\ell_\infty$ bounded attacker as it is against the 20-step PGD attack with a single random start we tested against previously. Note that DeepFool is not $\ell_\infty$ bounded and thus the perturbed images are outside of the robustness radius enforced during adversarial querying. Additional experiments on CIFAR-FS can be found in Tables 18, 19, 20.

| Model | $\mathcal{A}_{DF}$ | $\mathcal{A}_{MI}$ | $\mathcal{A}_{20-PGD}$ |
|---|---|---|---|
| R2-D2 | 7.91% | 0.01% | 0.0% |
| R2-D2 AQ (ours) | **14.45%** | **31.87%** | **30.31%** |
| R2-D2 Transfer | 0.42% | 24.01% | 19.75% |

Table 18: 5-shot MiniImageNet results against DeepFool (DF) (2 iteration) $\ell_\infty$ attack, MI-FGSM (MI) ($\epsilon = 8/255$) attack, and PGD attack with 20 random restarts (20-PGD). We compare R2-D2 trained with adversarial-querying (AQ) to the adversarially trained transfer learning R2-D2 as in section 4.1.

| Model | $\mathcal{A}_{DF}$ | $\mathcal{A}_{MI}$ | $\mathcal{A}_{20-PGD}$ |
|---|---|---|---|
| R2-D2 | 0.00% | 0.39% | 0.01% |
| R2-D2 AQ (ours) | **14.45%** | **53.46%** | **46.57%** |
| R2-D2 AT (Transfer Learning) | 1.41% | 38.28% | 33.17% |

Table 19: 5-shot CIFAR-FS results against DeepFool (DF) (2 iteration) $\ell_\infty$ attack, MI-FGSM (MI) ($\epsilon = 8/255$) attack, and PGD attack with 20 random restarts (20-PGD). We compare R2-D2 trained with adversarial-querying (AQ) to the transfer learning R2-D2 as in section 4.1.

| Model | $\mathcal{A}_{ResNet}$ |
|---|---|
| R2-D2 | 0.00% |
| R2-D2 AQ (ours) | **59.68%** |
| R2-D2 AT (Transfer Learning) | 42.02% |

Table 20: 5-shot CIFAR-FS results against black-box transfer attacks crafted on an adversarially trained (transfer learning) ResNet-12 model using 7-PGD. We then test R2-D2 trained with adversarial-querying (AQ) and the transfer learning R2-D2 model on these crafted perturbations.

## A.7 Experiments on heads vs. backbones

| Model | 1-shot MI | 5-shot MI | 1-shot FS | 5-shot FS |
|---|---|---|---|---|
| R2-D2 | **20.59%** | **31.52%** | **32.33%** | **44.80%** |
| MetaOptNet | 18.37% | 28.08% | 30.74% | 43.79% |
| MetaOptNet (R2-D2 backbone) | 18.81% | 24.68% | 29.57% | 41.90% |
| ProtoNet (R2-D2 backbone) | 18.24% | 28.39% | 26.48% | 40.59% |

Table 21: Robust test accuracy of adversarially queried R2-D2, MetaOptNet, and the MetaOptNet and heads with R2-D2 backbone on Mini-ImageNet (MI) CIFAR-FS (FS) datasets. Robust accuracy is computed with respect to a 20-step PGD attack.

| Model | 1-shot MI | 5-shot MI | 1-shot FS | 5-shot FS |
|---|---|---|---|---|
| R2-D2 | 55.22% | 73.02% | 68.36% | 82.81% |
| MetaOptNet | **60.65%** | **78.12%** | **70.99%** | **84.11%** |
| MetaOptNet (R2-D2 backbone) | 55.78% | 73.15% | 68.37% | 82.71% |

Table 22: Natural test accuracy of naturally trained R2-D2, MetaOptNet, and the MetaOptNet head with R2-D2 backbone on the Mini-ImageNet (MI) and CIFAR-FS (FS) data sets.

## A.8 Experiments on alternatives to adversarial training

| Model | $\mathcal{A}_{nat}$ | $\mathcal{A}_{adv}$ |
|---|---|---|
| R2-D2 | 73.02% | 0.00% |
| R2-D2 AQ | 57.87% | **31.52%** |
| R2-D2 AQ Denoising | 57.68% | 31.14% |

Table 23: 5-shot MiniImageNet results for our highest performing R2-D2 with feature denoising blocks. $\mathcal{A}_{nat}$ and $\mathcal{A}_{adv}$ are natural and robust test accuracy, respectively, where robust accuracy is computed with respect to a 20-step PGD attack. Top robust accuracy is bolded.

| Model | $\mathcal{A}_{nat}$ | $\mathcal{A}_{adv}$ |
|---|---|---|
| R2-D2 | 82.81% | 0.00% |
| R2-D2 AQ (ours) | 69.25% | **44.80%** |
| R2-D2 with SR defense | 35.15% | 23.00% |
| R2-D2 with DefenseGAN | 35.15% | 28.05% |

Table 24: 5-shot CIFAR-FS results comparing the superresolution defense (SR defense) and DefenseGAN. $\mathcal{A}_{nat}$ and $\mathcal{A}_{adv}$ are natural and robust test accuracy, respectively, where robust accuracy is computed with respect to a 20-step PGD attack. Both methods perform worse than their adversarially queried counterpart. Top robust accuracy is bolded.

## A.9 PGD attack

## A.10 Runtime Comparisons

In this section, we compare the training speeds of adversarially queried models to their naturally meta-learned counterparts. See Table 25.

| Model | Clean | AQ |
|---|---|---|
| ProtoNet | 0.0472 | 0.0650 |
| R2-D2 | 0.0672 | 0.18167 |
| MetaOptNet | 0.3256 | 1.5425 |

Table 25: Hours elapsed per epoch of training on 4 NVIDIA RTX 2080 Ti graphics cards.

---
**Algorithm 3** PGD Attack
---
**Require:** network, $F_\theta$, input data, $(\mathbf{x}, y)$, number of steps, $n$, step size, $\gamma$, and attack bound, $\epsilon$.
Initialize $\delta \in \mathcal{B}_\epsilon(\mathbf{x})$ randomly
**for** $i = 1, \ldots, n$ **do**
    Compute $g = \text{sign}\left(\nabla_\delta \mathcal{L}_\theta\left(\mathbf{x} + \delta, y\right)\right)$.
    Update $\delta = \delta + \gamma g$.
    **if** $\|\delta\|_p > \epsilon$ **then**
        Project $\delta$ onto the surface of $\mathcal{B}_\epsilon(\mathbf{x})$.
    **end if**
    **if** $\text{argmax}\, F_\theta(\mathbf{x} + \delta) \neq y$ **then**
        break
    **end if**
**end for**
**return** perturbed image $\mathbf{x} + \delta$
---