[Reviews · NeurIPS 2020]

Review 1

Summary and Contributions: The proposed work addresses the notion of model robustness in few shot regime. The authors show that naturally trained meta learners are not robust to adversarial examples and develop a robust meta-learner which uses adversarial examples on the query set to improve robustness.

Strengths: The authors develop a simple algorithm-agnostic method to improve robustness in the context of few shot learning and show that models trained using Adversarial Query (AQ) method are more robust than their naturally trained counterparts. Another interesting observation is that they show transfer learning from Adversarially trained model has lower performance than AQ. They also discuss the effectiveness of commonly used methods from adversarial examples literature such as preprocessing and denoising blocks in the context of few shot learning.

Weaknesses: Although the proposed approach is simple, some experimental findings can be explained better. From the results of Tables 7 and 8, it seems that perturbing support data does not provide any advantage (in the case of Omniglot dataset it is less robust than even the naturally trained model). Similarly, it is unclear why AQ is more robust than transfer learning from adversarially trained models. Previous works[1,2] have shown that transfer learning can be a very good baseline for few-shot classification so the significant reduction in natural accuracy is an interesting observation. [1] A Closer look at few shot classification, Chen et al., arXiv:1904.04232 [2] A Baseline for few shot classification, Dhillon et al., arXiv:1909.02729

Correctness: Yes

Clarity: The structure of the paper and writing quality of the paper can be improved. The usage of multiple small tables makes the paper little difficult to read and can make the results from different experimental settings difficult to summarize. The best results from the algorithm is seen in Table 9 which could have been included along with the main results.

Relation to Prior Work: Yes, the authors have highlighted how the proposed work is different from previous approaches.

Reproducibility: Yes

Additional Feedback: A few additional concerns: — Most of the experiments are done using the backbone from R2-D2. It would be interesting to see the effect of using a larger backbone on the transfer learning experiments. — The attack algorithms used for evaluation are based on l_\infty norm,does the robustness hold across different norms as well? — In Table 16 and Table 17, we expect adversarial accuracy to increase as 1/lambda increases. However, we see a drop in robustness. why? — In L164, it is mentioned that "Attacking only support data can be seen as maximizing clean test accuracy when fine-tuned in robust manner". Why?Its unclear why that should occur. To summarize, the proposed approach is interesting and could encourage further research towards robust models for few shot learning. However, the reasoning for some of the experimental findings is unclear and the paper needs to be restructured for easier understanding. ------ Update: The authors have addressed my concerns about the experiments with transfer learning and additional attack norm. The novelty of this work lies in perturbing only query data, but it seems natural that perturbing support data as well should increase robustness which is not observed. This concern was shared by other reviewers as well and more description/justification was required to explain this. For the META-Trades experiments as well, the original paper[30] showed that only natural accuracy decreases while robust accuracy improves when increasing 1/lambda. This is different from the networking behaving as a constant function. The authors are encouraged to look into this in more detail.


Review 2

Summary and Contributions: Summary: This work develops an approach to producing a meta-learner robust to adversarial examples in few-shot classification. The main difference (or say the novelty in this work), between the proposed algorithm in Algorithm 2 and the classical meta-learner in Algorithm 1, is that this work proposes to construct adversarial queries in the inner loop of meta-learning. The authors thoroughly investigate the causes for adversarial vulnerability. They also demonstrate the superior robust performance of the proposed method, on Mini-ImageNet and CIFAR-FS, to robust transfer learning for few-shot image classification. Contributions: 1. This paper proposes a new method called adversarial querying, in which adversarial queries are constructed in the inner loop of meta-learning. 2. This paper thoroughly investigates the causes for adversarial vulnerability of four meta-learning algorithms, which is quite impressive to me.

Strengths: 1. The proposed idea of constructing adversarial queries in the inner loop of meta-learning is simple, and good performance due to this idea is demonstrated. 2. The investigation of adversarial vulnerability of the four meta-learning algorithms is thorough. To me, this is the most impressive strength of this work.

Weaknesses: 1. This work is mostly empirical, as with many papers in this field; the results can be more convincing to me if any theoretical investigation can be provided, although I understand this may be hard. 2. Only four meta-learning algorithms (MAML, R2-D2, MetaOptNet, and ProtoNet) are tested with the proposed idea; more state-of-the-art meta-learning algorithms can be considered.

Correctness: To me, the conclusions drawn from empirical results are reasonable, and the empirical studies are thorough and impressive.

Clarity: The subsections in section 4 and section 5 look divergent to me for some reason, but I think overall the paper is well written.

Relation to Prior Work: The paper explains the relationship with related work including ADML in [29], a robust meta-learner to adversarial attacks based on MAML. If a more detailed comparison between the algorithm of ADML with the proposed Algorithm 2 can be provided, it will be more clearly about the difference between these two adversarial meta-learners, as well as the novelty of the proposed method, but I understand the limit of space.

Reproducibility: Yes

Additional Feedback: ######## After reading other reviewers' comments and the authors' responses, I decide to lower down my score, for the concerns about the novelty and the empirical investigation of this work.


Review 3

Summary and Contributions: This paper address the issue that few-shot learning methods are highly vulnerable to adversarial examples. The authors show that naturally trained meta-learning methods are not robust by PGD attack. Then the authors propose adversarial querying to solve the issue. The authors find that meta-learning models are most robust when the feature extractor is fixed, and only the last layer is retrained during fine-tuning. They also show that choice of classification head significantly impacts robustness.

Strengths: - The problem this paper studied is important. - This paper is the first one to propose a meta-learning approach for robust few-shot learning, which is novel. - This paper compares AQ with transfer learning form adversarially trained models, and find AQ is better. - This paper further study the robustness-accuracy trade-off, only fine-tuning the last layer, the R2-D2, etc. In summary, I think this paper well study the robustness of few-shot learning and deeply investigate the AQ method.

Weaknesses: The main algorithm is somewhat simple. It is just an adversarial training, using a meta-learning framework which is standard for few-shot learning. My concern is the novelty of the algorithm. But the problem is interesting and this paper moves the first step for developing adversarially robust methods for few-shot applications.

Correctness: Yes. I haven't seen anything wrong so far, but I am not the expert in this area.

Clarity: I suggest the authors can show at least one visual case instead of just accuracy numbers to better illustrate the problem and your performance.

Relation to Prior Work: Yes. This work is the first one to use meta-learning to achieve robustness for few-shot learning. It is clear that the method of this work is novel and different from previous works.

Reproducibility: Yes

Additional Feedback:


Review 4

Summary and Contributions: The paper proposes to make few-shot learning on image classification tasks: MiniImageNet, Omniglot and CIFAR-FS robust to PGD based adversarial attacks. To this end, the authors propose a simple solution: to optimise the loss in the outer loop of any meta-learning algorithm over adversarially perturbed query data points. The authors provide an extensive empirical evaluation using 4 baseline meta-learning algorithms: MAML, ProtoNet, R2-D2 and MetaOptNet and compare the performance of their method of adversarial querying across different configurations. Their results show that AQ does improve the adversarial robustness to PGD attacks for few-shot learning tasks.

Strengths: 1) To the best of my knowledge, this is the first work considering the problem of adversarial robustness of few-shot learning in image classification tasks. This work can prove to be an important starting point for robustness research for few-shot learning. 2) I really like the initial evaluation to support the claim of why existing methods are not very robust to PGD attacks, this provides motivation towards developing better meta-learning algorithms. Also their empirical comparison of meta-learning approaches being more robust than transfer learning approaches in Table 6 is insightful. 3) The empirical results do indicate that adversarial querying improves the robustness of few-shot learning on MiniImageNet, Omniglot and CIFAR-FS.

Weaknesses: [Edit after Author Response] I thank the authors for acknowledging the suggestion for merging the tables, captioning and moving Algorithm 1 to the Appendix. 1) Discussing the findings of Table 7 and 8 and providing a reasoning for them is fairly important since this detail forms the crux of your simple idea. The author response does not elaborate or explain too much on this, but rather states the observations from Table 8. 2) While I thank the authors for performing more experiments on state of the art meta-learning approaches like MCT and mentioning that AQ on MCT reduces the drop of natural accuracy, the current results in the paper using other meta-learning approaches do have a large drop in the natural accuracy. This certainly diminishes the practical use of AQ. ------- I agree that the paper does have some good positive points. However I am slightly inclined towards a rejection currently primarily due to the following reasons: 1) The core idea of this paper is very simple and straightforward. Though the authors justify that they are the first to do it, I am unsure whether this work might count as a novel enough contribution for the NeurIPS community. 2) While it is common knowledge that adversarial training leads to a drop in the natural accuracy while improving the adversarial robustness accuracy, the drop is usually small and not very significant. In contrast, from results in Table 4,5 vs Table 2, it appears that using AQ causes a big drop (sometimes almost 15-20%) in the natural accuracy. While the adversarial robustness increases, such a big drop in natural accuracy is not supportive of their AQ technique being very useful practically. 3) Apart from providing some empirical results in Table 7, 8, the authors provide very little reasoning/exploration of why only perturbing the query data and not the support data or both the (query+support data) is used in their proposed technique. This seems to be the crux of their simple idea, but is not well justified. I would have liked some better intuitive explanation and possibly some theoretical justification as to why only fine-tuning in the outer-loop with adversarially perturbed query data is sufficient to make the model robust to previously unseen few-shot (support, query) pairs. 4) Instead of having so many small individual tables, I would rather have a single big table comparing different methods/experiments (especially since almost all the tables are over the same 2 datasets). There are 2 main reasons for this: a) First, having separate tables makes it difficult to compare the improvements of a technique with respect to other techniques whose results have been put in a separate table. b) There are several repeating results across different tables which seems to be unnecessarily used for space filling.

Correctness: Yes, the methods used by the authors along with the empirical evaluation is correct, though lacking justification in places.

Clarity: For the most part the paper is well written. The tables however are sometimes tough to comprehend, especially since none of the table captions describe the brief findings from the results. Further, in some places the tables are not referenced from within the text and hence it becomes slightly confusing as to which tables contain which results (especially tables 7,8).

Relation to Prior Work: The paper clearly discusses prior work and related work of few-shot learning and adversarial attacks and learning.

Reproducibility: Yes

Additional Feedback: Questions: 1) I am surprised at the big difference in A_{adv} values of "MAML adv. query" and "MAML adv. query and support" in Table 8. Can you provide any intuition/justification of this observation? Suggestions: 1) The table captions are very poorly indicative. A majority of the table titles contain the exact common strings which are not necessary. Instead, since there are so many tables, you should indicate in 1 line what the table is trying to evaluate, and a 1 line description in words of the findings from the table's results. You don't need to repeat everywhere that nat and adv refers to natural and robustness accuracy (these can be mentioned just once in the first table). 2) The basic description of the meta-learning algorithm (Algorithm 1) consumes too much space without adding anything novel and should be shifted to the Appendix according to me.


Review 5

Summary and Contributions: The paper studies performance of meta-learning algorithms for few-shot learning under white/black-box adversarial attacks. Systematic experiments with adversarial learning (mainly with projected gradient attacks -- PGD) lead the authors to propose to train meta-learners with adversarially modified query data (constructed with PGD). The proposed method shows favorable results with several meta-learners (MAML, ProtoNet, R2-D2, MetaOptNet) on a few datasets (Mini-ImageNet, Omniglot, and CIFAR-FS) under white-box attacks. Furthermore, favorable results are also demonstrated when compared to architectural and pre-processing based defenses.

Strengths: The proposed approach is intuitive and simple to implement. It can be used with any episodically trained, meta-learner. The proposed approach is motivated by systematic experiments. Additional experiments with black box attacks are presented. Comparisons are made with a wide range of other defenses.

Weaknesses: The proposed approach is somewhat obvious, it is an almost out-of-the-box application of [17] to the episodic training. In this sense, the contributions of this paper are mostly to provide baseline empirical results.

Correctness: Questions/Issues: 1-The proposed adversarially trained approach performs considerably worse when no adversarial attack is present (57.87% vs 73.01%), why is that? Since an attacker can choose to attack or simply not change the input, a reasonable performance measure is the average performance (A_{nat} + A_{adv}) / 2. Which yields 44.69% on 5-shot ImageNet compared to the baseline (R2-D2) 36.51% -- still demonstrating the advantage of the proposed method, though not as great as comparing A_{adv} alone. 2-The proposed approach was not demonstrated on meta-learners that minimize a *support* set loss during meta-training. For example, instead of minimizing the generalization error one can minimize the leave-one-out validation error (e.g. I believe REPTILE does this). It is not clear if this method would apply in that setting, it would be nice to see an experiment on this. 3-Results in each section are not necessarily complete. Certain sections present results on some datasets (and shots) and not others. For example: section 4.2 presents results on only 5-shot ImageNet, supporting evidence on doing query-only augmentation could be made stronger with results on other datasets. Also same section: MAML 5-shot on mini ImageNet should be 63% and not the reported 60%. Same with the experiment in Section 4.1 -- where are the analogous experiments on other datasets? 4-Finally it should be noted that the datasets used in this paper are somewhat small and do not test out-of-distribution generalization. I would like to see results on other datasets (e.g. MetaDataset) which contains far more data and tests out-of-distribution generalization.

Clarity: The paper is mostly well written. A few minor points: 1-copyright-control is not a safety critical application. 2-section 2.3 needs more clarification is the term "retraining" is used to mean "fine-tuning" are the layers in question re-initialized to random? Line 97: " against a weak attacker" -- what kind of attack? How weak?

Relation to Prior Work: As far as the paper mentions the only prior work on adversarial meta-learning was [29]. I am not aware of other works myself.

Reproducibility: Yes

Additional Feedback:

[Author Response · NeurIPS 2020]

We thank the reviewers for their insightful comments. A number of additional experiments were suggested. We
have completed each of these experiments, and we have added them to our current version. We first enumerate these
experiments and then respond to individual reviewer questions. The details of these experiments are as follows:

**A1.** Transfer learning with a deeper backbone: We have now tested transfer learning with an adversarially trained
ResNet-50. The deeper backbone actually decreases robust accuracy on few-shot tasks compared to the much shallower
R2-D2 backbone (15.63% vs. 17.72% on 1-shot mini-ImageNet). We have updated our paper to include these results.
**A2.** Additional attack norm: We have now implemented the $\ell_2$ PGD following Madry, and we find that AQ models
are also robust to these attacks. For example, our adversarially queried R2-D2 model achieves 35.53% robust 5-shot
accuracy on mini-ImageNet, while the analogous transfer learned model achieves only 15.92% on the same task.
**A3.** Out-of-distribution testing: We have now evaluated our models on Meta-Dataset. For example, on the FGVC-
Aircraft dataset, R2-D2 AQ gets 36.65% 5-shot robust accuracy, while the analogous transfer learned model gets
20.83%. For the same models, on the CUB-200 dataset, the robust accuracies are 29.04% and 20.05%, respectively.
**A4.** Additional meta-learning algorithms: We have now implemented and run adversarial querying with the state-of-the-
art meta-learning method, MCT. This experiment yielded high robustness and small natural accuracy tradeoff. MCT
(AQ) simultaneously achieves 79.9% natural accuracy and 53.2% robust accuracy on 5-shot mini-ImageNet.
**A5.** Reptile: We have trained an adversarial Reptile variant, and we are updating our paper to include these experiments.

**Reviewer 1:**
*"Multiple small tables makes the paper difficult to read… Table 9 could have been included along with the main results."*
Thank you for your suggestions. We have merged tables and highlighted these results in our current version.
*"It seems that perturbing support data does not provide any advantage."* Thank you for pointing this out. In Section 4.2,
we note that perturbing support data optimizes the network for adversarial fine-tuning. Likewise, in Table 8, we see that
models trained with adversarial support achieve better robustness than the naturally trained model (still worse than AQ
models) only when adversarially fine-tuned at test time. We have added a subsection to better explain this phenomenon.
*"It is unclear why AQ is more robust than transfer learning."* Ding et al. 2019 found that unlike natural accuracy,
robustness does not generalize well under transfer learning. AQ produces models specifically optimized for robust
few-shot adaptation. We have updated our paper to discuss this issue in detail.
*"Attacking only support data can be seen as maximizing clean test accuracy when fine-tuned in a robust manner"*
The meta-objective in this case is clean query loss after optimization on adversarial support data. This optimization
simulates natural performance at test-time after fine-tuning on adversarial few-shot data.
*"We expect adversarial accuracy to increase as $\frac{1}{\lambda}$ increases."* Thank you for pointing this out. When $\frac{1}{\lambda}$ is sufficiently
large, the network is encouraged to behave as a constant function and to neglect accuracy (both natural and robust).
Consider that a network need not make correct predictions to achieve low KL divergence loss.
**Reviewer 2:**
*"Only four meta-learning algorithms are tested."* We have now tested adversarial querying on the state-of-the-art MCT
method, and we have updated our paper accordingly. See A4 above for additional details.
**Reviewer 3:**
*"Show at least one visual case."* We agree that visualizations could make our work easier to understand. We have now
both visualized adversarial examples and constructed a visual representation of our algorithm.
**Reviewer 4:**
Regarding your suggestions concerning table aggregation and captioning, table referencing, and moving Algorithm 1 to
the Appendix, we agree with your assessment, and we have updated our current version to reflect these changes.
*"I am surprised at the difference in $A_{adv}$ values of "MAML adv. query" and "MAML adv. query and support"…*
*perturbing the query data and not the support data"* Thank you for pointing this out. In Section 4.2, we note that
perturbing support data optimizes the network for adversarial fine-tuning. Likewise, in Table 8, we see that models
trained with adversarial support achieve better robustness than the naturally trained model (still worse than AQ models)
but only when adversarially fine-tuned at test time. We have updated our explanation of this phenomenon in Section 4.2.
*"AQ causes a big drop [in natural accuracy]"* This massive trade-off exists in the standard setting where SOTA robust
ImageNet models achieve 65.30% clean accuracy while clean-trained models achieve 88.50%. We agree that this is an
obstacle for deployment. In our new MCT tests (see A4 above), we see a far smaller drop of only 6%.
**Reviewer 5:**
*"…results on some datasets and not others. For example: section 4.2 presents results on only 5-shot ImageNet."* The
appendix in our submitted version contains these experiments. We agree that these experiments are important.
*"…meta-learners that minimize \*support\* set loss."* Thank you, we have now included your suggestion (see A5 above).
*"test out-of-distribution generalization."* We have now evaluated our models on Meta-Dataset (see A3 above).
*"A few minor points..."* Thank you for pointing these out. We have added clarification to our current draft.
*"…considerably worse when no attack is present."* This massive trade-off exists in the standard setting where SOTA
robust ImageNet models achieve 65.30% accuracy while clean-trained models achieve 88.50%. We agree that this is an
obstacle for deployment. We like your average performance metric and have updated our current version to include this.

[Meta-Review · NeurIPS 2020]

The submission proposes a method called adversarial querying (AQ) to tackle the problem of adversarial robustness in few-shot learning. Adversarial querying works by applying an adversarial perturbation to the query set when meta-training in an effort to find a few-shot learner parameterization which is robust to adversarial attacks when tuned on the support set of a given learning problem. Results in the paper show that naturally trained few-shot learners are very sensitive to adversarial attacks. Adversarial robustness results are presented for a variety of benchmarks (mini-ImageNet, CIFAR-FS, Omniglot) and learners (Prototypical Networks, R2-D2, MetaOptNet, MAML). The proposed approach is shown to yield better adversarial robustness than competing approaches (transfer learning from an adversarially-trained backbone, ADML) while maintaining a better clean accuracy. Strengths identified by reviewers include the algorithm-agnostic nature of the proposed approach and the systematic nature of the empirical investigation (e.g. experiments with many meta-learners and adversarial attacks, comparison against a transfer learning baseline). Several questions raised by reviewers were satisfyingly addressed in the rebuttal through additional experiments (Reviewer 1: transfer learning with a deeper backbone and robustness across different norms; Reviewer 2: testing with state-of-the-art meta-learning algorithms; Reviewer 4: applying AQ to Reptile and testing on Meta-Dataset). Multiple reviewers noted that the proposed approach is a very straightforward application of adversarial training to the meta-learning setting, and as such its technical contribution is somewhat thin. While experiments are extensive, they do not present conclusions that are surprising or challenge existing beliefs, which means that the submission’s contribution is judged mainly from an empirical perspective. Reviewers 4 and 5 were concerned with the large gap in natural accuracy between AQ and its non-adversarially-trained baseline. Authors agreed in the rebuttal that this is an obstacle for deployment and point out that such an obstacle also exists in the standard adversarial robustness setting. I think it would be unfair to ask of the submission that it completely overcomes this obstacle, but it’s fair to ask that it acknowledges and discusses the natural accuracy gap, especially in the context of few-shot classification, where the small number of labeled examples already contributes to an accuracy degradation. Given that the authors have already taken action on many of the reviewers’ suggestions (for instance via additional experiments), I am fairly confident that they would follow through on this suggestion if the submission was accepted. Several reviewers also expressed concerns with the paper’s justification for perturbing only the query set. The authors justify this choice by pointing out that the two alternatives either do not target the right objective (support-only perturbations) or do not offer empirical benefits over adversarial querying (support+query perturbations; Tables 7, 8, 14, and 15). Reviewers find that the empirical results are counter-intuitive and lack a clear explanation. I agree that this observation is puzzling, and that providing a convincing explanation would greatly strengthen the submission (although I would note that the observation in itself is still a valuable contribution). What needs to be determined is whether the paper has enough merits as it is to justify acceptance. While there’s clear room for improvement, I think the NeurIPS community would still benefit from the submission due to its extensive empirical evaluation. I therefore recommend acceptance.